# Characterization of RNA interference in the cnidarian *Nematostella vectensis* reveals partial target silencing but lack of small RNA amplification

Yael Admoni[1]*, Magda Lewandowska[1], Reuven Aharoni[1], Junchao Shi[2¤], Xudong Zhang[2], Qi Chen[2], Yehu Moran [1]*

1 Department of Ecology, Evolution and Behavior, Alexander Silberman Institute of Life Sciences, Faculty of Science, Hebrew University of Jerusalem, Jerusalem, Israel, 2 Molecular Medicine Program, Department of Human Genetics, and Division of Urology, Department of Surgery, School of Medicine, University of Utah, Salt Lake City, Utah, United States of America

¤ Current address: China National Centre for Bioinformation and Beijing Institute of Genomics, Chinese Academy of Sciences, Beijing, China
* yael.admoni@mail.huji.ac.il (YA); yehu.moran@mail.huji.ac.il (YM)

## Abstract

RNA interference (RNAi) is a sequence-specific mRNA degradation mechanism, in which short interfering RNAs (siRNAs) guide Argonaute proteins to complementary targets, resulting in their degradation. In many organisms, RNAi also serves antiviral roles by processing viral double-stranded RNA (dsRNA) into siRNAs that prevent viral replication. Antiviral RNAi is considered an ancestral mechanism which invertebrates rely on for defense against viruses, whereas vertebrates have evolved instead the interferon pathway. Recent studies suggest that sea anemones, members of the basally-branching phylum Cnidaria, might possess an innate immune response with more vertebrate characteristics than previously thought; however, it is unknown whether cnidarians also employ RNAi as an antiviral response similarly to nematodes and insects. Here, we characterize the response of the model cnidarian *Nematostella vectensis* to simulated viral infection. We injected dsRNA with eGFP sequence into eGFP-expressing transgenic zygotes and show that siRNAs mapping to the eGFP sequence are generated and induce a moderate but significant knockdown of eGFP expression. Interestingly, we detected no evidence for secondary siRNA production, despite their crucial role in the amplification of antiviral response in other organisms. Notably, siRNA pathway components are specifically upregulated upon dsRNA injection, while microRNA pathway components are downregulated. Furthermore, injection of mRNA coding for self-replicating viral gene fused to eGFP, also induced upregulation of siRNA-related genes and a mild decrease in transgene expression. Overall, we propose that *N. vectensis* possesses an siRNA-mediated response that lacks secondary amplification and likely functions as a short-term antiviral mechanism.

**Data availability statement:** All new small RNA sequencing data in this paper are publicly available on the SRA repository under identification number PRJNA1145188 https://www.ncbi.nlm.nih.gov/bioproject/?term=PRJNA1145188.

**Funding:** This work was supported by European Research Council Consolidator Grant 863809 to YM. https://cordis.europa.eu/project/id/863809 The funders did not play any role in the study design, data collection and analysis, decision to publish, or preparation of the manuscript.

**Competing interests:** The authors have declared that no competing interests exist.

**Abbreviations:** AGO, Argonaute; AlkB, α-ketoglutarate-dependent hydroxylase; dsRNA, double-stranded RNA; eGFP, enhanced GFP; FHV, Flock House Virus; hpi, hours post-injection; IP, immunoprecipitation; IRFs, Interferon Regulatory Factors; miRNA, microRNA; nt, nucleotide; PANDORA-seq, Panoramic RNA display by overcoming RNA modification aborted sequencing; RdRps, RNA-dependent RNA polymerases; RISC, RNA-Induced Silencing Complex; RNAi, RNA interference; RppH, RNA 5′ Pyrophosphohydrolase; RT-qPCR, reverse transcription-quantitative PCR; siRNA, short interfering RNA; srGFP, scrambled sequence of eGFP; T4PNK, T4 polynucleotide kinase; WT, wild-type.

## Introduction

RNA interference (RNAi) is a conserved gene expression regulation mechanism that encompasses several pathways mediated by 20–30 nucleotide (nt) long RNA molecules, which silence specific targets through base pairing. The three major RNAi-related pathways are the microRNA (miRNA), short interfering RNA (siRNA), and PIWI-interacting RNA. Among these, the miRNA pathway primarily targets endogenous protein-coding genes and the siRNA pathway provides defense against transposons and viral infection [1,2]. The siRNA pathway is considered to be the main antiviral immune mechanism of invertebrates, based on its ancestral origin and studies of common invertebrate models such as the nematode *Caenorhabditis elegans* and the fly *Drosophila melanogaster* [3–5]. In vertebrates, the siRNA system persists only in specific tissues or developmental stages, and in general their main antiviral innate immune response instead relies heavily on the interferon pathway [6–9]. The siRNA mechanism is based on detection of double-stranded RNA (dsRNA) by cellular receptors that leads to processing of the dsRNA by RNase III Dicer into 20–25 bases long primary siRNAs [10–12]. These siRNAs are loaded into Argonaute (AGO) proteins, integral components of the RNA-Induced Silencing Complex (RISC), that carry them to the target viral RNA and apply mechanisms to prevent its expression [13]. As an important step, the primary response is amplified by the production of secondary siRNAs that greatly increase target silencing and prolong the response. In nematodes and plants, RNA-dependent RNA polymerases (RdRps) are responsible for secondary siRNA production [14–16], whereas in flies and mosquitoes, the amplification is RdRp-independent and is mediated by reverse transcriptases [17,18]. In *C. elegans*, secondary siRNAs are typified by carrying 5′-triphosphate modification and they are 22 nt long and tend to display sequence biases [19–22]. *D. melanogaster* also generates secondary antiviral siRNAs with 5′-triphosphate modifications, likely synthesized by an as-yet-unidentified reverse transcriptase [18,23]. Antiviral RNAi is well described in nematodes, insects, and plants; however, major gaps remain in its evolutionary history as it has not been characterized in many lineages, including early branching metazoans, especially as both insects and nematodes belong to the Ecdysozoa.

In this study, we investigate the antiviral immune response in the sea anemone *Nematostella vectensis*, a model organism representing the phylum Cnidaria that diverged from most other animals over 600 million years ago [24,25]. Cnidarians, comprising the Anthozoa (sea anemones and corals) and the Medusozoa (jellyfish and hydras), occupy a key phylogenetic position for exploring the early evolution of immune pathways [26]. Intriguingly, despite their basal position, some cnidarians exhibit unexpected complexity in their antiviral pathways [27,28], and so far, there has been contradictory evidence regarding the functionality of the antiviral RNAi system in cnidarians.

First, *N. vectensis* possesses homologs to antiviral RNAi-related components such as AGO2, Dicer1, and three RdRps that were shown to be upregulated upon injection of the viral dsRNA mimic, polyinosinic:polycytidylic acid [poly(I:C)] [27]. This could

point to activation of the antiviral siRNA pathway; however, *N. vectensis* also has homologs to components of the interferon pathway, such as Interferon Regulatory Factors (IRFs) [27–29]. In *N. vectensis*, interferon-related components are upregulated following poly(I:C) injection along with RNAi-related genes and it is unclear if one of these pathways predominates in the antiviral response [27].

Second, all cnidarians employ a miRNA system, hence possess homologs to components of the antiviral RNAi system such as AGO and Dicer. In *N. vectensis*, the two AGO paralogs are specialized in carrying distinct populations of small RNAs, with AGO1 loading only miRNAs and AGO2 loading both miRNAs and endogenous siRNAs [30,31]. Evidence for RNAi activity has also been reported in other cnidarians. In the medusozoan *Hydra magnipapillata*, dsRNA-driven RNAi has been successfully used to achieve gene knockdown by electroporation or feeding [32,33]. In the sea anemone *Exaiptasia diaphana* (formerly *Aiptasia pallida*), dsRNA was shown to be effective when delivered by soaking [34]; however, no further studies reported the successful employment of this method in *Exaiptasia* or other sea anemones in the last 17 years. In addition, no siRNAs corresponding to viral sequences were found in the sea anemone *Actinia equina* [35], and failures to perform gene knockdown by dsRNA were reported in *N. vectensis* [36], pointing against robust activity of the antiviral RNAi pathway.

To address the contradicting evidence and clarify whether antiviral RNAi is an immune mechanism in cnidarians, we characterized the molecular response to dsRNA presence in *N. vectensis* zygotes. Because no natural viruses have yet been isolated from *N. vectensis*, we used synthetic approaches to mimic viral infection by injecting dsRNA into zygotes. This was complemented by injection of mRNA coding for the Flock House Virus (FHV) RdRp. Similar methods have been employed in *Drosophila* and *C. elegans* to simulate viral replication and activate antiviral RNAi [37–39].

## Results

### Injection of dsRNA results in mild and transient transgene knockdown

To test the activation of the antiviral RNAi pathway in *N. vectensis*, we utilized a transgenic line of anemones that ubiquitously expresses enhanced GFP (eGFP) under the actin promoter [40)]. We injected dsRNA carrying the sequence of eGFP into individual heterozygous zygotes of this line (Fig 1A) and visualized the embryos under a fluorescence stereomicroscope. As control, we injected dsRNA with a scrambled sequence of eGFP (srGFP). After 24 hours, the fluorescence of eGFP was dimmer in the embryos injected with eGFP dsRNA compared to srGFP (Fig 1B). We continued to track the fluorescence for 72 hours and documented that the difference in eGFP fluorescence between the injected groups is no longer visually distinguishable at this time point (Fig 1C). To validate this at the molecular level, we performed reverse transcription-quantitative PCR (RT-qPCR) and confirmed that at 24 hours post-injection (hpi) eGFP transcripts are significantly lower in eGFP dsRNA injected animals; however, the level of knockdown is relatively modest, only 40% compared to srGFP injected animals (Fig 1D and S1 Data). As expected, no significant difference in eGFP transcript level was detected by RT-qPCR at 72 hpi (Fig 1D and S1 Data). In addition, we quantified the fluorescence intensity of eGFP from stereomicroscope images and showed that animals injected with eGFP dsRNA exhibited significantly lower mean fluorescence intensity compared to those injected with srGFP, although the difference was modest, consistent with the moderate reduction observed at the transcript level (Fig 1E and S1 Data). By 72 hpi, fluorescence intensity was indistinguishable between treatments, in accordance with the visual and RT-qPCR results (Fig 1E and S1 Data).

### dsRNA injection into *N. vectensis* zygotes leads to generation of siRNAs

The knockdown of eGFP following dsRNA injection suggested that a sequence-specific small RNA silencing pathway was activated; hence, we proceeded with investigating if siRNAs, which are the hallmark of antiviral RNAi in nematodes, flies, and plants, were produced. To detect eGFP-derived siRNAs, we generated small RNA libraries for both *actin::eGFP* heterozygotes and wild-type (WT) animals injected with eGFP dsRNA, at 24 hours hpi. We mapped the 18–30 nt long

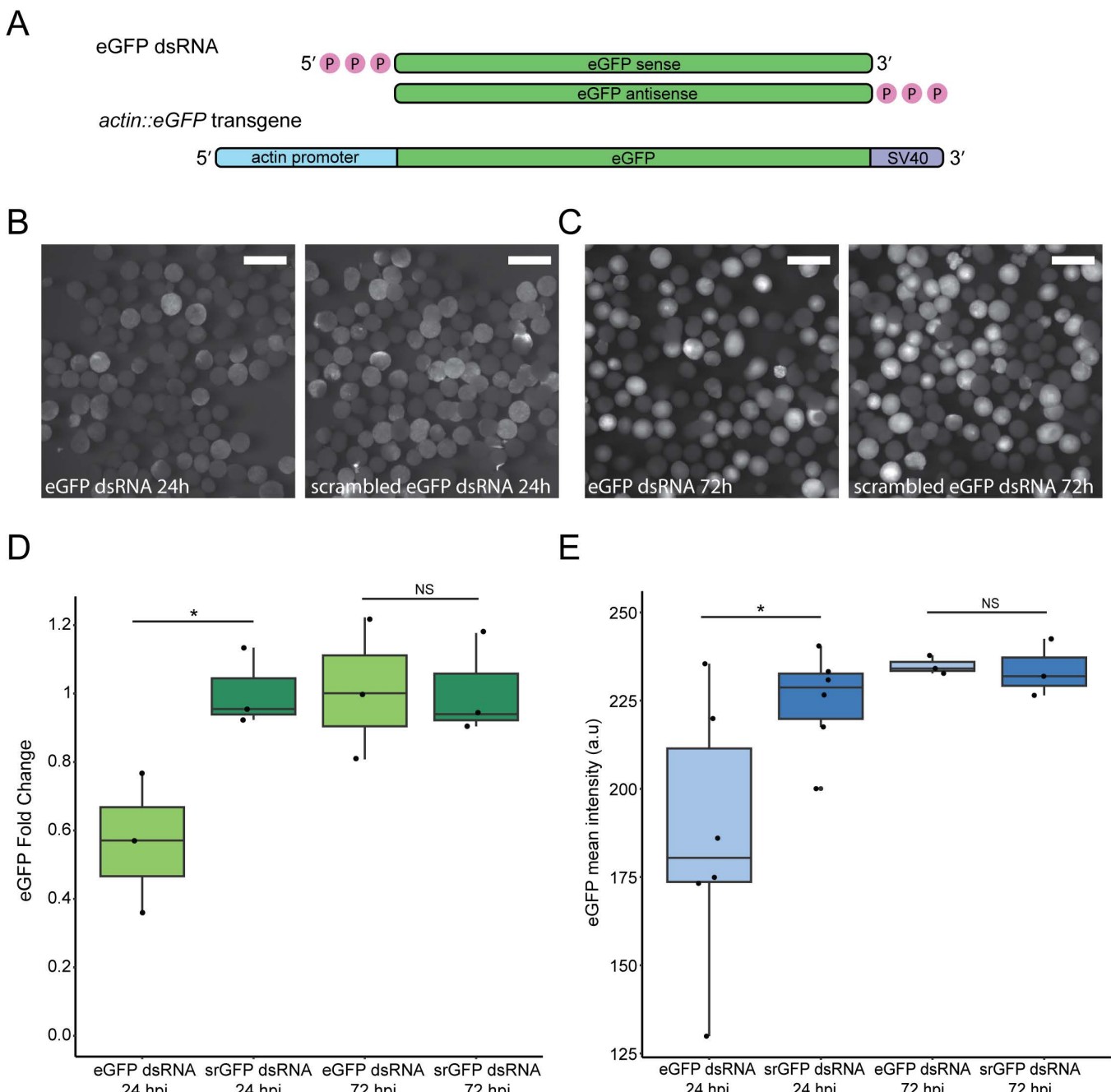

**Fig 1. dsRNA injection to *N. vectensis* zygotes triggers mild and transient gene knockdown. (A)** Schematic representation of the dsRNA carrying eGFP sequence used in this study and of the transgene of the *actin::eGFP* line [40]. **(B)** Heterozygous *actin::eGFP* embryos showing dimmer eGFP fluorescence 24 hpi of dsRNA carrying eGFP sequence (left) compared to scrambled eGFP sequence (right). Scale bars represent 500 μm. **(C)** No difference in eGFP fluorescence of heterozygous *actin::eGFP* embryos is exhibited 72 hpi of dsRNA carrying eGFP sequence (left) compared to scrambled eGFP sequence (right). Scale bars represent 500 μm. **(D)** eGFP transcript levels fold change measured by reverse transcription-quantitative PCR (RT-qPCR), 24 and 72 hpi of dsRNA. Statistical significance is shown for pairwise comparison (one-tailed Student *t* test, *n* = 3 biological replicates, p-values = 0.02, 0.48). **(E)** eGFP mean intensity values, measured at 24 and 72 hpi of dsRNA. Statistical significance is shown for pairwise comparison (one-tailed Student *t* test, *n* = 6 biological replicates for 24 hpi and *n* = 3 for 72 hpi, *p*-values = 0.02, 0.59). Box plots indicate the median and interquartile range, with whiskers showing the minimum and maximum values. The data underlying this Figure can be found in S1 Data.

small RNA reads to eGFP sequence and revealed a significant enrichment of reads mapping to eGFP in groups injected with dsRNA compared to the mock injected controls (Fig 2A and S1 Data). For this analysis, read counts were normalized to spikeins for quantitative comparison of small RNA abundance across samples (Fig 2) and in addition calculated in RPM (S1A and S1B Fig). Consistent with the difference in read counts, eGFP fluorescence was visibly weaker in dsRNA-injected *actin::eGFP* heterozygotes compared to controls, and the eGFP transcript levels were measured lower by RT-qPCR (S2A and S2B Fig). As expected, there was no difference in read counts that map to eGFP between the *actin::eGFP* and the WT animals (Fig 2A). Moreover, the size distribution of the reads peaked at 20–21 nt long, which resembles the length of siRNA in other organisms but 1 nt shorter and is in accordance with endogenous siRNAs previously reported for *N. vectensis* [12,19,41] (Fig 2B). We then visualized the mapping locations to eGFP and interestingly found a reoccurring pattern between all replicates in both *actin::eGFP* and WT injected animals (Fig 2C and 2D and S1 Data). This indicates that reads mapping to eGFP are not a result of random degradation of the injected dsRNA but of mechanistic processing. The origin of the reads from the sense or antisense strand of the dsRNA also showed a recurring pattern between replicates (Fig 2C and 2D and S1 Data).

Next, we analyzed the phasing of the siRNA reads, which can indicate whether the dsRNA is processed by Dicer in a slowly processive manner from one side of the sequence or that processing is initiated by cleavage at internal sites [42,43]. The structure of the injected dsRNA allows processing from the ends (Fig 1A). We tested for both 20 and 21 nt registers, corresponding to the two dominant size peaks (Fig 2B), for transgenic and WT datasets. In all cases, the radar plots showed no pronounced periodicity (S3A–S3D Fig), indicating a lack of strong phasing. We also examined nucleotide composition biases, since some siRNA types in frequently studied organisms like flies and nematodes exhibit 5′ nt preferences [44–46]. Analysis of both total (S3E and S3F Fig) and uniquely mapping reads (S3G and S3H Fig) showed no notable sequence bias for either the 20 or 21 nt siRNA populations.

We then proceeded to validate that the eGFP-derived siRNAs are loaded into the AGOs, the carriers of siRNAs within the RISC. For this purpose, we collected WT zygotes injected with eGFP dsRNA at 24 hpi and applied immunoprecipitation (IP) of AGO2 using specific antibodies, followed by small RNA sequencing. The read count of eGFP-mapped siRNAs was significantly higher in AGO2 IP compared to IgG control, confirming the loading of the dsRNA-derived siRNAs (S4A Fig and S1 Data). The size distribution of the reads repeated the 20–21 nt peaks seen in the non-IP libraries (Figs 2B and S4B). Furthermore, we employed Trans-kingdom rapid affordable Purification of RISCs (TraPR), a method that purifies RISCs and their associated small RNAs from diverse taxa [47]. Corroborating the AGO2 IP results, in WT animals injected with dsRNA, the read counts of eGFP-mapping siRNAs were significantly more abundant than in uninjected animals (S4C Fig and S1 Data). The size distribution followed a similar pattern of 20–21 nt peaks (S4D Fig). To assess the efficiency of the loading of dsRNA-derived siRNAs, we compared the number of reads to the total reads of miRNAs captured by the two AGO-isolation methods, since miRNAs represent efficiently loaded molecules. For both AGO2 IP and TraPR methods, eGFP-mapping siRNAs are captured in significantly higher numbers compared to miRNAs (S4E and S4F Fig, respectively, S1 Data), implying that the dsRNA-derived siRNAs are loaded with high efficiency into AGO2. Together, these results demonstrate that injected dsRNA is processed in *N. vectensis* into 20–21 nt siRNAs that are incorporated into AGO2.

## No production of secondary siRNAs was detected in *N. vectensis*

Secondary siRNAs production is a major part of antiviral RNAi response in *C. elegans* and *Drosophila*, by amplifying the response and enhancing the magnitude and duration of viral RNA silencing. Therefore, we wished to test if this process also occurs in *N. vectensis*. Since *N. vectensis* has four RdRps, three of which are upregulated upon viral mimic injection [27], we hypothesized that secondary siRNAs might be produced by RdRps, hence contain a 5′-triphosphate moiety, similarly to *C. elegans* secondary siRNAs [19]. This modification prevents detection by 5′-ligation-dependent sequencing; thus, we applied treatment by RNA 5′ Pyrophosphohydrolase (RppH) to remove the modification and expose secondary siRNAs to ligation

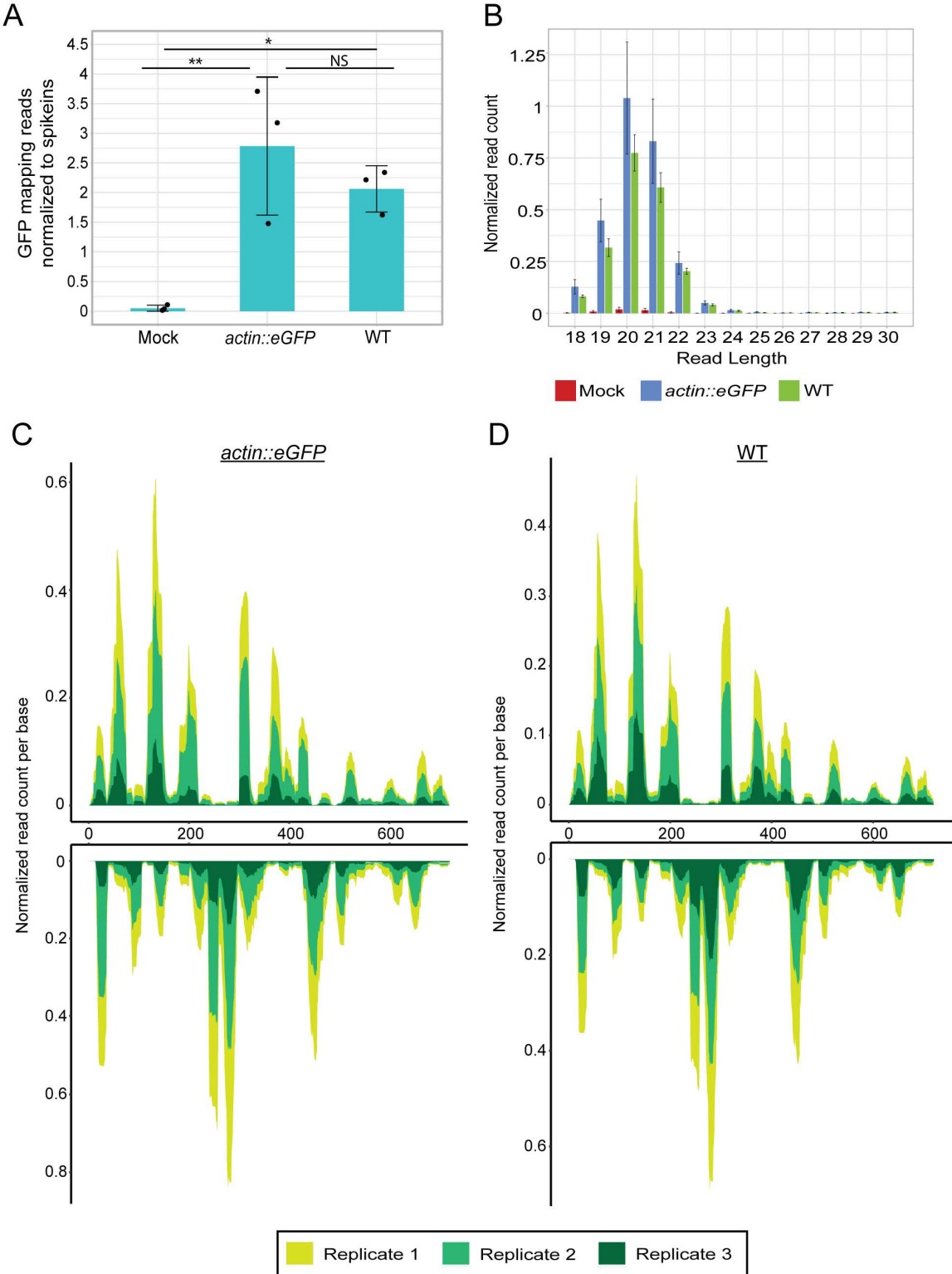

**Fig 2. Primary siRNAs are generated following dsRNA injection. (A)** Normalized number of reads mapping to eGFP sequence, from embryos injected with dsRNA carrying eGFP sequence (24 hpi). *Actin::eGFP* zygotes were injected with either dsRNA or mock injection mix. WT zygotes were injected with dsRNA. Statistical significance is shown for pairwise comparison (one-way ANOVA, *n* = 3 biological replicates, *p*-value = 0.008, pairwise

comparison with Tukey's HSD post-hoc test, *p*-values = 0.007, 0.03, 0.47). **(B)** Size distribution of reads mapping to eGFP sequence corresponding to the samples in (A). **(C, D)** eGFP sequence coverage by small RNA reads from *actin::eGFP* (C) or WT (D) embryos injected with eGFP dsRNA. The top graph represents the sense strand, and the bottom graph represents the antisense strand. Biological replicates are represented by different colors. Bar graphs show the mean; error bars represent the standard deviation in (A) and the standard error in (B). The data underlying this Figure can be found in S1 Data.

during small RNA libraries preparation. The treatment was applied to the same RNA extracted from *actin::eGFP* and WT embryos that were injected with eGFP dsRNA and was used for the small RNA libraries in Fig 2. The small RNA sequencing results revealed no significant difference in read counts mapping to eGFP between the RppH-treated and untreated libraries (Fig 3A and S1 Data). The reads were normalized to spikeins (Fig 3) and calculated in RPM (S1C Fig). In addition, there was no difference in the size distribution of the reads between the treatments, unlike similar experiments in *C. elegans* (Fig 3B) [19,20,22]. These results concluded that no secondary siRNAs that chemically resemble the ones in *C. elegans* were generated. Nonetheless, unlike in nematodes and flies, most plants secondary siRNAs do not include 5′-triphosphate, suggesting that the absence of this modification cannot rule out by itself the production of secondary siRNAs [16].

To investigate this further, we took a different approach, based on a mechanism that occurs in *C. elegans*, in which secondary siRNAs are generated using the target mRNA as template for RNA-dependent RNA replication [19,48]. To test if this mechanism is active in *N. vectensis*, we injected into *actin::eGFP* transgenic zygotes a shorter eGFP dsRNA, which includes only the 3′ half of the original sequence, and looked for siRNAs that map upstream to it, covering the rest of the eGFP sequence. The presence of siRNAs mapping upstream of 3′ half would suggest that the eGFP mRNA transcribed from the transgene is used as template for the transcription of the 5′-mapping siRNAs. Following sequencing of small RNAs from half-eGFP compared to full-length-eGFP dsRNA injected animals, it was apparent that siRNA reads map only to the 3′ half of the eGFP (Fig 3C and S1 Data). The injection of half-eGFP was still able to knock down eGFP levels, as eGFP transcript levels were not significantly different between half-eGFP and full-length-eGFP injected animals, and fluorescence was visibly similar (Figs 3D and S5A and S1 Data). Moreover, there was no difference between the number and size distribution of reads mapping to eGFP (S5B, S5C, and S1D Figs and S1 Data). Consistent with previous sequencing results, we observed a recurring mapping pattern between replicates (Fig 3C and S1 Data).

After learning that secondary siRNAs are not generated by similar mechanisms to *C. elegans*, we tested whether any species of siRNAs could be detected by removing a variety of RNA modifications, by performing Panoramic RNA display by overcoming RNA modification aborted sequencing (PANDORA-seq) that combines treatments by T4 polynucleotide kinase and α-ketoglutarate-dependent hydroxylase and was shown to expose numerous types of modified small RNAs to sequencing [49]. PANDORA-seq is able to remove RNA methylations such as $m^1A$, $m^3C$, $m^1G$, and $m^2_2G$ and convert 3′ phosphate and 2′3′ cyclic phosphate into 3′ hydroxyl, which allows sequencing of modified classes of RNAs that are hidden from traditional sequencing. For this assay, four replicates of *actin::eGFP* were injected with dsRNA and collected at 24 hpi, and the extracted RNA was used for PANDORA-seq and traditional small RNA sequencing for comparison. Mapping the reads from both PANDORA-seq and traditional sequencing to eGFP sequence did not reveal an increase in reads mapping to eGFP or any major changes in mapping distribution (Fig 3E and 3F and S1 Data). The mapping peaks differ in read count, which could be explained by the fact that PANDORA-seq detects a larger variety of small RNAs, hence decreasing the representation of dsRNA-originating siRNAs in the libraries. The size distribution of the reads mapping to eGFP also did not reveal a difference in siRNAs following the removal of modifications (Fig 3G and 3H).

As reported in other systems, a prerequisite step in the production of secondary siRNAs in flies and mosquitoes is reverse transcription of the viral dsRNA [17,18]. To test whether a similar process could be occurring in *N. vectensis*, we analyzed genomic DNA from embryos collected 72 hpi with eGFP dsRNA using PCR with eGFP-specific primers. No amplification of the eGFP sequence was detected, whereas the endogenous control gene was successfully amplified (S6A and S6B Fig), indicating that the injected dsRNA was not reverse transcribed in the anemone embryos. To verify that

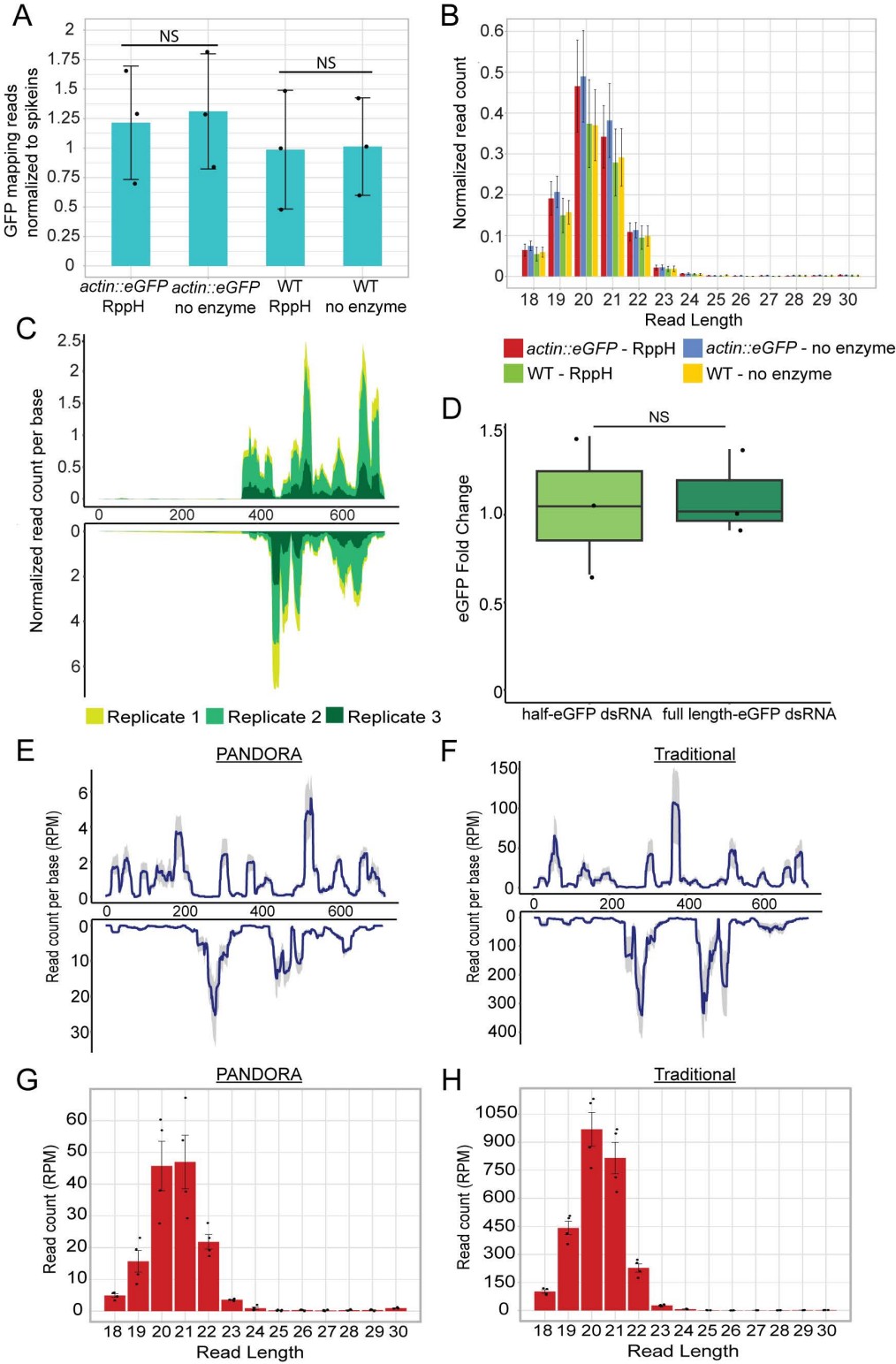

**Fig 3. No secondary siRNAs were detected following dsRNA injection. (A)** Normalized number of reads mapping to eGFP sequence following treatment of RNA with RppH enzyme or mock treatment without enzyme. The treatment was performed on RNA extracted from *actin::eGFP* or WT embryos injected with dsRNA carrying eGFP sequence (24 hpi). Statistical significance is shown for pairwise comparison (one-way ANOVA, *n* = 3 biological

replicates, *p*-value = 0.8, pairwise comparison with Tukey's HSD post-hoc test, *p*-values = 0.99, 0.99). **(B)** Size distribution of reads mapping to eGFP sequence corresponding to the samples in (A). **(C)** eGFP sequence coverage by small RNA reads from *actin::eGFP* embryos injected with 3′ half-eGFP dsRNA (24 hpi). The top graph represents the sense strand, and the bottom graph represents the antisense strand. Biological replicates are represented by different colors. **(D)** eGFP transcript levels fold change measured by RT-qPCR, 24 hpi of either half-dsRNA or full-length-dsRNA. Statistical significance is shown for pairwise comparison (one-tailed Student *t* test, *n* = 3 biological replicates, *p*-value = 0.38). **(E, F)** eGFP sequence coverage by small RNA reads sequenced by PANDORA-seq (E) or traditional small RNA sequencing (F). RNA was extracted from *actin::eGFP* embryos injected with dsRNA carrying eGFP sequence (24 hpi). The top graph represents the sense strand, and the bottom graph represents the antisense strand. The blue line represents the mean of *n* = 4 biological replicates and the grey is the confidence interval. **(G)** Size distribution of reads mapping to eGFP sequence corresponding to the samples in (E). **(H)** Size distribution of reads mapping to eGFP sequence corresponding to the samples in (F). Bar graphs show the mean; error bars represent the standard deviation in (A) and the standard error in (B) and (G, H). Box plot indicates the median and interquartile range, with whiskers showing the minimum and maximum values. The data underlying this Figure can be found in S1 Data.

the absence of eGFP silencing at 72 hpi was not due to degradation of the injected dsRNA, we extracted total RNA from injected animals and performed PCR using cDNA and the same set of primers. eGFP amplicons were detected at both 24 and 72 hpi, (S6C and S6D Fig), demonstrating that the dsRNA remained present during this period.

## Injection of viral sequence induces upregulation of siRNA components and reduction of eGFP expression

To the best of our knowledge, viruses from cnidarian hosts were never isolated or cultured. Hence, to generate a mimic of viral infection that is not just dsRNA, we synthesized mRNA encoding the FHV RNA1 segment, which contains the viral RdRp, fused to eGFP (hereafter referred to as FHV-eGFP mRNA). When expressed in flies and nematodes, FHV RNA1 triggers antiviral RNAi by self-replicating [37–39]. We hypothesized that expression of the FHV-eGFP construct would initiate self-replication, leading to the formation of dsRNA intermediates containing eGFP sequences that could activate the siRNA-mediated antiviral response and reduce eGFP transgene expression. To test this hypothesis, we injected FHV-eGFP mRNA into *actin::eGFP* zygotes and collected embryos at 24 and 48 hpi. As a negative control, we injected mRNA encoding mCherry protein, which is not expected to trigger antiviral activity. Fluorescence microscopy revealed no clear difference in eGFP intensity at 24 hpi, but a visibly weaker signal in FHV-eGFP injected animals at 48 hpi compared to mCherry injected controls (Fig 4A and 4B). Quantification of mean fluorescence from captured images confirmed a significant yet modest reduction in eGFP fluorescence at both time points (Fig 4C and 4D), indicating a mild but significant knockdown of eGFP expression consistent with the response observed following direct dsRNA injection (Fig 1E). Next, we measured the transcript levels of eGFP and the siRNA-related genes *AGO2* and *Dicer1* by RT-qPCR. At 24 hpi, eGFP transcript levels appeared elevated in FHV-eGFP-injected embryos relative to controls, likely reflecting the presence of injected mRNA (Fig 4E). By 48 hpi, this difference was no longer apparent, and eGFP expression showed a mild, non-significant decrease (Fig 4F), possibly due to siRNA-mediated targeting of the replication-derived dsRNA. Notably, both *AGO2* and *Dicer1* were strongly upregulated at 24 hpi, indicating activation of the siRNA pathway in response to likely replication of the viral RNA sequence (Fig 4E). This upregulation diminished by 48 hpi (Fig 4F), similar to the transient immune activation pattern previously observed in *N. vectensis* following poly(I:C) treatment [27]. Together, these results suggest that expression of the FHV-eGFP construct triggered a transient siRNA-mediated response in *N. vectensis*, leading to measurable downregulation of eGFP expression.

## dsRNA injection leads to downregulation of miRNA system components

*N. vectensis* has a functional miRNA system which employs components such as Dicer1, AGO1, and AGO2 that were shown to be crucial for the early development of this species [31,50,51]. It was shown that AGO1 specifically carries miR-NAs and AGO2 carries both miRNAs and endogenous siRNAs [31]. Another miRNA system component is GW182 (called TNRC6 in vertebrates) that has a homolog in *N. vectensis* and was shown to mediate gene silencing when expressed heterologously in mammalian cells [52]. Previous results revealed that injection of poly(I:C), a dsRNA mimic, into *N. vectensis*

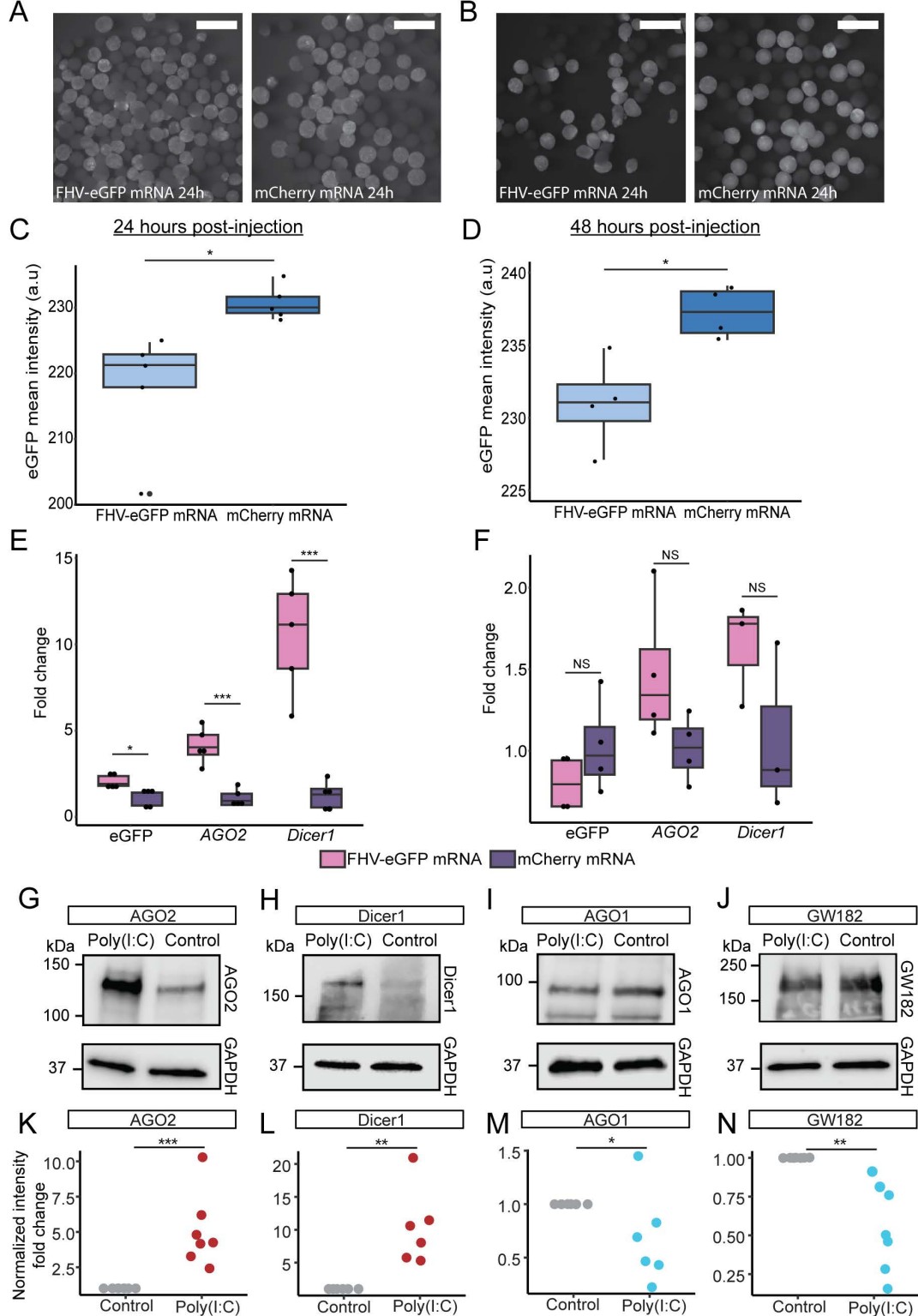

**Fig 4. Activation of the siRNA pathway by FHV-eGFP mRNA and upon poly(I:C) treatment. (A)** Heterozygous *actin::eGFP* embryos 24 hpi of FHV-eGFP mRNA coding for viral RdRp fused to eGFP (left) or mCherry mRNA as negative control (right). Scale bars represent 500 μm. **(B)** Heterozygous *actin::eGFP* embryos 48 hpi of FHV-eGFP mRNA coding for viral RdRp fused to eGFP (left) or mCherry mRNA as negative control (right). Scale bars

represent 500 μm. **(C)** eGFP mean intensity values, measured at 24 hpi of FHV-eGFP or mCherry mRNA. Statistical significance is shown for pairwise comparison (two-tailed Student $t$ test, $n = 5$ biological replicates, p-values = 0.01). **(D)** eGFP mean intensity values, measured at 48 hpi of FHV-eGFP or mCherry mRNA. Statistical significance is shown for pairwise comparison (two-tailed Student $t$ test, $n = 4$ biological replicates, p-values = 0.01). **(E)** Fold change of eGFP, *AGO2*, and *Dicer1* measured by RT-qPCR, 24 hpi of FHV-eGFP or mCherry mRNA. Statistical significance is shown for pairwise comparison (two-tailed Student $t$ test, $n = 5$ biological replicates, p-value = 0.02261 for eGFP, and p-values < 0.001 for *AGO2*, and *Dicer1*. **(F)** Fold change of eGFP, *AGO2*, and *Dicer1* measured by RT-qPCR, 48 hpi of FHV-eGFP or mCherry mRNA. Statistical significance is shown for pairwise comparison (two-tailed Student t *t*est, $n = 4$ biological replicates, p-values = 0.22, 0.08, 0.17). **(G–J)** Western blots of antiviral RNAi-related components AGO2 (G) and Dicer1 (H) and miRNA system specific components AGO1 (I) and GW182 (J) performed on protein content extracted from WT embryos, 48 hpi of poly(I:C) or NaCl control (protein sizes: AGO2 122 kDa, Dicer1 171 kDa, AGO1 96 kDa, and GW182 185 kDa). **(K–N)** Fold change of normalized band intensity corresponding to the Western blots in (G–J), showing RNAi-related components signal in response to poly(I:C) or NaCl control. Statistical significance is shown for pairwise comparison (two-tailed Student $t$ test or Mann–Whitney–Wilcoxon Test, $n \geq 6$ biological replicates, p-value < 0.001 for AGO2, and p-values = 0.004, 0.02, 0.001, for Dicer1, AGO1, and GW182, respectively). Box plots indicate the median and interquartile range, with whiskers showing the minimum and maximum values. The data underlying this Figure can be found in S1 Data.

zygotes leads to upregulation of *AGO2* and *Dicer1*, but no change of *AGO1* or *GW182* on the RNA level, a result that phenocopies the injection of dsRNA [27]. To further characterize the activation of the RNAi pathways in *N. vectensis* we performed western blot using RNAi components-specific antibodies following injection of poly(I:C) to zygotes. In accordance with previous results, the western blot demonstrates upregulation of AGO2 and Dicer1 (Fig 4G and 4H). Interestingly, AGO1 and GW182 that are miRNA-specific components, seem to be mildly downregulated (Fig 4I and 4J). This was found to be statistically significant by comparison of normalized band intensities (Fig 4K–4N and S1 Data). To assess the effect of the antiviral immune response on miRNA expression levels, we analyzed miRNA expression after eGFP dsRNA injection. The total read count of all *N. vectensis* miRNAs was not affected (S7A Fig and S1 Data). A total of 10–12 miRNAs were differentially expressed, with no clear trend regarding the function of their target genes (S7B and S7C Fig).

Recently, single-cell RNA sequencing datasets from *N. vectensis* identified immune cell clusters in both embryos and adults [53,54]. In embryos, the immune cell cluster was defined based on immune activation following injection of the viral mimic poly(I:C) into zygotes of an RLRb reporter line [54]. Within this immune cell population, both *Dicer1* and *AGO2* exhibited enrichment, with average $\log_2$-fold change of ~4 and ~3, respectively (S8 Fig and Table 1). In contrast, *AGO1* does not show enrichment. This observation is consistent with the upregulation of AGO2 and Dicer1 detected by western blot analysis (Fig 4K and 4L) and implies that this upregulation occurs mostly in specialized immune cells.

## Discussion

Antiviral RNAi is an ancestral pathway that provides crucial defense against viruses to nematodes and insects, both belonging to the superphylum Ecdysozoa. The contradictory evidence regarding its activity in cnidarians has motivated us to test whether it has a role in the sea anemone *Nematostella vectensis*. Using eGFP dsRNA injection to zygotes of *actin::eGFP* anemones, we show that dsRNA-driven, likely antiviral RNAi response is activated, and it results in a significant but mild silencing of eGFP expression, at the RNA and fluorescence level (Fig 1). This comes in contrast to the striking effect of silencing by RNAi exhibited in other organisms [39,55]. Nonetheless, we show that siRNAs are generated

**Table 1. Gene expression in the immune cell cluster.**

| Gene name | Cell type | Average $\log_2$-fold change | Adjusted *p*-value | Percentage of immune cells expressing the gene | Percentage of non-immune cells expressing the gene |
|---|---|---|---|---|---|
| *Dicer1* | Immune | 4.049 | 0 | 0.381 | 0.029 |
| *AGO2* | Immune | 3.032 | 1.25E-118 | 0.116 | 0.015 |
| *AGO1* | Immune | 0.1809 | 1 | 0.079 | 0.069 |

Expression of *Dicer1*, *AGO2*, and *AGO1*, analyzed from single-cell RNA-seq data [54] obtained from poly(I:C) treated RLRb reporter line.

by processing of dsRNA into 20–21 nt long small RNAs, in a sequence repetitive manner across all replicates (Fig 2). The recurrence of sense- or antisense-specific mapping of each small RNA resembles the preference of AGO to select one of the strands of the small RNA duplex it loads. This phenomenon is known from the miRNA pathway, where the guide strand gets selected by AGO, whereas the star (also called passenger strand) is ejected and eventually degraded, resulting in much higher representation of the guide strand in sequencing data [56,57]. The highly reproducible mapping pattern of the siRNAs could suggest that Dicer has a sequence preference for processing the injected dsRNA, as seen in previous experiments in mammalian cells [8]; but it is more likely a result of AGO strand-selection of stochastically processed siRNAs, as no siRNA phasing was observed (S3A–S3D Fig). Supporting the functionality of the eGFP-derived siRNAs is the efficient loading of the eGFP-mapping siRNAs into AGO2, the carrier of siRNAs in *N. vectensis* (S4 Fig).

The eGFP-derived siRNAs do not show phasing, indicating that dsRNA processing likely occurs at multiple internal sites rather than progressively from one end, as seen in virally infected mouse cells [42]. This processing pattern resembles transfection of mouse cells with dsRNA with long single-strand overhang, suggesting that in *N. vectensis* Dicer processing starts by internal cleavage even when the dsRNA ends are accessible [43]. This observation is consistent with the dispersed distribution of siRNA reads across the eGFP sequence (Fig 2C and 2D). It was previously reported that Dicer1 in *N. vectensis* contains a DEAD-box helicase domain that highly resembles this domain in dicer proteins from other organisms [30]. This domain is generally associated with the ability of Dicer proteins to engage and process long dsRNA [58], and its conservation in *N. vectensis* Dicer1 is consistent with a capacity to process dsRNA.

Previous studies demonstrated that injection of long dsRNA in *N. vectensis* triggers upregulation of immune-related genes, including *AGO2* and *Dicer1* within 24 hours, followed by a decline at 48 hours [27]. In this study, we show that injection of mRNA encoding the FHV RdRp (fused to eGFP) similarly induces strong upregulation of the siRNA machinery, peaking at 24 hpi (Fig 4E and 4F). This response is driven by RdRp expression, as it is absent in mCherry RNA-injected animals, but since self-replication was not directly tested in this study, the formation of dsRNA intermediates cannot be definitively stated. The activation of the siRNA pathway was accompanied by a modest, yet significant, reduction in eGFP fluorescence, mirroring the effect observed following direct dsRNA injection (Fig 4C and 4D). This effect is likely, a result of FHV-eGFP-derived siRNA generation. Together, these results suggest that both dsRNA exposure and expression of a viral RdRp can activate the siRNA-mediated response in *N. vectensis*, resulting in siRNA production, as demonstrated for dsRNA injection, and transient downregulation of eGFP expression. We note that *N. vectensis* or any other cnidarian currently lacks cultured viruses suitable for experimental infection assays; therefore, dsRNA and viral mRNA were used as functional mimics of viral replication. Future studies using direct viral infections will be essential for more definitive validation of antiviral activity in *N. vectensis*; however, considering the robust upregulation of immune-related genes such as *RLR*s and *IRF* along with *AGO2* and *Dicer1* in response to dsRNA and poly(I:C) trigger [27], the enrichment of siRNA pathway components in immune cell clusters (S8 Fig), the specific activation of siRNA in contrast to miRNA pathway (Fig 4K–4N) and the characteristics of the siRNA pathway documented here, our findings could support the existence of a transient siRNA-mediated antiviral defense mechanism in *N. vectensis*.

The short-lived and mild knockdown effect caused by dsRNA injection could be explained by the lack of secondary siRNA generation. Three different methods were used in this study to detect secondary siRNAs, resulting in no detection, as well as no reverse transcription of the injected dsRNA (Figs 3, S6A, and S6B). Hypothetically, there could be secondary siRNAs in *N. vectensis* carrying yet unidentified modifications masking them from sequencing; however, the short-term knockdown effect strongly supports the notion that there is no secondary amplification of the RNAi response. This result correlates with the moderate knockdown effect and suggests that the siRNA-mediated and likely antiviral response in *N. vectensis* is not amplified by secondary siRNA generation, unlike in nematodes, flies, and plants. It can also explain the lack of siRNAs carrying viral sequences from natural sea anemone populations [35]. Another factor that might contribute to the moderate effect is that apoptosis is activated in *N. vectensis* upon poly(I:C) injection to zygotes [59]. Apoptosis is a mechanism of host defense against viruses and its increased activity at 24 hpi could mean that a stronger RNAi response

is not feasible or crucial in *N. vectensis*. It was shown that in mammalian cells the RNAi response is inhibited in cells that carry out the interferon response, and inhibition of the interferon pathway reveals a functional RNAi response in differentiated mammalian cells [60,61]. In *N. vectensis*, homologs of canonical vertebrate antiviral factors that take part in the interferon response are upregulated in response to poly(I:C) [27]. Unlike in mammals where interferon and RNAi were shown to be mostly antagonistic towards each other [62,63], knockdown of the dsRNA receptor RLRb, a homolog of the mammalian MDA5, in *N. vectensis* in combination with poly(I:C) injection leads to downregulation of *AGO2* and *Dicer1* [27], which hints that the interferon-like and RNAi pathways act in a synergetic manner in this species. In addition, it is likely that this synergetic relationship also exists in adults of *N. vectensis* and not limited to the embryos, which we have tested in this work. An overlap of differentially expressed genes was observed between poly(I:C) triggered embryos and 2′3′-Cyclic GMP-AMP triggered juvenile polyps [28]. Furthermore, similar gene markers were reported to be expressed in immune cell clusters in adults and in embryos [53,54]. This suggests a similar immune response pattern between embryos and adults, implying that the dsRNA-driven RNAi response in adults also lacks secondary amplification and has a transient role in immune defense.

RNAi, especially in its form as homogenously-processed miRNAs, is an extremely useful tool to regulate gene expression levels, which could explain its wide conservation, even in organisms where alternative antiviral pathways have evolved [8,9]. miRNAs tend to provide milder and more accurate regulation of expression levels compared to siRNAs and hence are less suitable for antiviral roles [64]. Interestingly, the *N. vectensis* components specifically associated with the miRNA pathway (AGO1 and GW182) are mildly downregulated at the protein level under dsRNA challenge, whereas those that serve both miRNAs and siRNAs (Dicer1 and AGO2) are strongly upregulated (Fig 4K–4N). This pattern supports the idea that the siRNA pathway plays a role in the response to invasive dsRNA, while transient downregulation of the miRNA machinery may serve to prioritize the likely-antiviral siRNA activity during infection.

A few cases of loss of the RNAi system have been described in fungi, but these are considered to have occurred recently in response to viral mechanisms that adapted to hijack the RNAi system and could have harmful effects in the long run [65,66]. To fully understand the conservation of the antiviral RNAi system it must be characterized in other organisms from different phylogenetic lineages [67]. Based on previous findings and the current study in *N. vectensis*, it could be concluded that RNAi as an antiviral response might be conserved as a helpful immune mechanism, even if it is not the main one. Furthermore, our work suggests that the synthesis of secondary siRNAs and the resulting long-term silencing by RNAi, might have been lost in sea anemones, as it is found in insects, nematodes, and plants. Altogether, this work challenges the common view that antiviral RNAi is the main immune mechanism of invertebrates in general and calls for further research of this system in other lineages.

## Methods

### Sea anemone culture and spawning

*N. vectensis* polyps were grown in the dark at 18 °C in 16 ‰ artificial seawater and fed three times a week with freshly hatched *Artemia salina* nauplii. Spawning and fertilization were conducted as previously described [51,68]. *Actin::eGFP* anemones were kindly received from Dr. Aissam Ikmi (European Molecular Biology Laboratory, Heidelberg, Germany) [40]. For injection to *actin::eGFP* line, sperm from heterozygous *actin::eGFP* males was used to fertilize WT eggs.

### Microinjection of dsRNA

Microinjection to zygotes was performed with Eclipse Ti-S Inverted Research Microscopes (Nikon, Japan) connected to an Intensilight fiber fluorescent illumination system (Nikon) for visualization of the fluorescent injected mixture. The system is mounted with a NT88-V3 Micromanipulator Systems (Narishige, Japan). dsRNA carrying eGFP sequence (720 bp for full-length and 360 bp for half-length) was ordered from RiboPro (the Netherlands) and Synbio Technologies (USA) and was

aliquoted and kept in −80 °C until use. The dsRNA included 5′-triphosphate and blunt ends and was prepared by the companies by annealing of single strands followed by column purification. All injection mixes included dextran Alexa Fluor 594 at 50 ng/μl (Thermo Fisher Scientific, USA) as fluorescent tracer. The volume injected to each individual zygote is determined visually by the fluorescent tracer and is estimated as ~10 pl (although could be variable) [69]. dsRNA-injected concentration was chosen according to survival of the injected animals, which was very low (i.e., below 50%) above 7 ng/μl. For small RNA libraries eGFP dsRNA (RiboPro) was injected in concentration of 1.75 or 3.5 ng/μl, which is roughly estimated as 1.75e−05 and 3.5 e−05 ng, into zygotes of *actin::eGFP* line or WT, which are 200–300 μm in diameter [70]. Biological replicates of the same experiment were injected with the same concentration. The mock injection control included only water and tracer. Half-eGFP dsRNA was injected at the same molarity as full-length dsRNA. For RT-qPCR, eGFP dsRNA and srGFP (Synbio technologies) were injected at 7 ng/μl, which is roughly estimated as 7e−05 ng. For western blot, high molecular weight poly(I:C) (average size of 1.5–8 kb) in 0.9% NaCl (Invivogen, USA) was injected to WT zygotes in concentration of 6.25 ng/μl with 0.9% NaCl as control. Injected animals (hundreds) were kept in an incubator at 22 °C, counted and transferred to fresh 16 ‰ artificial seawater every day. The animals were visualized before being flash-frozen in liquid nitrogen and were kept in −80 °C until RNA, DNA, or protein extraction. All sequences used in this study appear in S2 Data.

## RNA extraction

For generating small RNA libraries, total RNA was extracted from a few hundred injected animals using TRIzol (Thermo Fisher Scientific) according to the manufacturer's protocol, with minor changes. 1 μl of RNA-grade glycogen (Roche, Switzerland) was added into the isopropanol during the isolation step and centrifugation for isolation and 75% ethanol wash was at 21,130 *g*. Final RNA pellets were re-suspended in 15–20 μl of RNase-free water (Merck Millipore, USA). RNA concentration was measured by Qubit RNA BR Assay Kit (Thermo Fisher Scientific). At this stage, most of the extracted RNA was kept in −80 °C until small RNA libraries preparation and 400 ng were separated to use for RT-qPCR. The separated RNA was treated with Turbo DNase twice for 30 min at 37 °C (Thermo Fisher Scientific) to remove residual genomic DNA followed by another round of purification. RNA integrity was assessed by gel electrophoresis with 1:1 formamide (Merck Millipore) and 1 μl of loading dye on 1.5% agarose gel for RT-qPCR and by Bioanalyzer Nanochip (Agilent, USA) for small RNA libraries. RNA that was not used for libraries was extracted using Quick-RNA microprep kit (Zymo Research, USA) with two rounds including Turbo DNase treatment.

## Reverse transcription-quantitative PCR (RT-qPCR)

eGFP transcripts were amplified by RT-qPCR using primers designed via Primer3 version 0.4.0 [71], Primer calibration showed product specificity and 115% efficiency, −2.99 slope and R2 > 0.98. AGO2 and Dicer1 primers are previously published [27] (primer sequences are available in S2 Data). cDNA was prepared using iScript according to manufacturer's protocol (Bio-Rad, USA) and was diluted to 5 ng/μl and stored at −20 °C. RT-qPCR was performed using StepOnePlus Real-Time PCR System v2.2 (ABI, Thermo Fisher Scientific) with Fast SYBR Green Master Mix (Thermo Fisher Scientific). Each sample was quantified in triplicates for the transcript of interest and housekeeping gene 4 (HKG4) as an internal control [72]. 5 ng of cDNA template was used for each technical replicate. The Reaction thermal profile was 95 °C for 20 s, then 40 amplification cycles of 95 °C for 3 s and 60 °C for 30 s, a dissociation cycle of 95 °C for 15 s and 60 °C for 1 min and then brought back to 95 °C for 15 s (+0.6 °C steps). Expression fold-change was analyzed using a comparative Ct method ($2^{-\Delta\Delta Ct}$) [73]. All RT-qPCR experiments included three biological replicates.

## Small RNA library preparation

Between 0.5 and 1 μg of extracted RNA was mixed with spikeins (lin-4-5p, miR-125a-5p, miR148a-3p and miR-659-5p, mixed to 0.025 fmol per 1 μg of RNA) and size selected on 15% urea-PAGE gel (Bio-Rad). RNA in the size of 18–30 nts was cut and extracted from gel by incubation overnight in 810 μl 0.3 M NaCl at 4 °C in rotation. Size-selected small

RNAs were precipitated at −20 °C for a minimum of 3 hours in 1 ml ice-cold 100% ethanol with 1 μl glycogen (Roche), then washed with 900 μl of ice-cold 75% ethanol, centrifuged at 21,130 *g* and resuspended in 7.5 μl RNase-free water. sRNAs extracted from AGO2 immunoprecipitated or TraPR isolated samples were not further processed. Next, small RNA libraries were prepared using NEBNext Multiplex Small RNA Library Prep Set for Illumina kit (New England Biolabs, USA) with some modifications [74]. Library preparation kit adapters 3′ and 5′ and reverse transcription primer were diluted 1:1 of stock concentration before use. For overnight ligation of 3′ adapter, T4 RNA Ligase 2 truncated KQ and Murine RNase inhibitor were used (New England Biolabs). The ligated products were subjected to 13–18 cycles of PCR amplification then Certified Low Range Ultra Agarose (Bio-Rad) stained with orange gel loading dye (New England Biolabs). Bands ranging between 137 and 149 nts (corresponding to adapter ligated small RNAs) were cut from gel and purified using NucleoSpin Gel and PCR Clean-up (Macherey-Nagel, Germany) with elution volume of 12 μl. Final concentrations were measured using Qubit HS dsDNA assay kit (Thermo Fisher Scientific). The quality of cDNA libraries was checked by using TapeStation system on High Sensitivity D1000 ScreenTape (Agilent). The libraries were validated for a dominant peak at the size range of 150–167 nts and sequenced with NextSeq500 (Illumina) in single end mode.

### Bioinformatic analysis

Small RNA reads were analyzed using miRDeep2 [75]. Sequences shorter than 18 nts were not included. Filtered reads were mapped to *N. vectensis* genome using Bowtie1 (version 1.3.1) [76]. Reads not mapped to the genome were then mapped to eGFP sequence. miRDeep2 quantifier.pl module was used with default parameters for quantification of miRNA reads. For miRNAs, differential expression edgeR was used [77]. For comparing miRNA total counts only miRNA guides that had at least two reads in one of the replicates were retained. Read counts were normalized to averaged spikeins reads of each sample or calculated in RPM for read count comparison. The normalized reads were averaged between three biological replicates. Phasing analysis was performed using coordinate-sorted eGFP-mapped reads according to the method described in [42,43]. Unique, perfectly-matching 20–22 nts long reads were divided into phases in modulo 20 and 21 separately. Representation of nucleotide composition was done using WebLogo [78], for reads of three biological replicates of dsRNA injected *actin::eGFP* zygotes.

### RNA 5′ Pyrophosphohydrolase (RppH) treatment

To remove potential 5′-triphosphate modification from small RNAs, total RNA was divided to two reactions, one was treated with RppH enzyme (New England Biolabs) according to manufacturer's protocol and the control libraries were treated with no enzyme.

### PANDORA-seq

PANDORA-seq was performed as described before [49]. Briefly, total extracted RNAs were separated through a 15% urea polyacrylamide gel and small RNAs of 15–50 nt were visualized with SYBR Gold solution (Invitrogen, USA). Fifteen–50 nt small RNAs were excised and recovered by gel elution buffer as previously described [79]. A sample of the eluted RNA was stored in −80 °C for traditional-seq, the remaining eluted RNA was then treated with T4 polynucleotide kinase (T4PNK) enzyme in a 50 μl reaction mixture (5 μl 10×PNK buffer, 1mM ATP, 10 U T4PNK) followed by RNA isolation with TRIzol (Thermo Fisher Scientific). The collected RNAs were then treated with α-ketoglutarate-dependent hydroxy-lase (AlkB) in a 50 μl reaction mixture (50mM HEPES, 75 μM ferrous ammonium sulfate, 1mM α-ketoglutaric acid, 2mM sodium ascorbate, 50mg/L BSA, 4 μg/mL AlkB, 2,000 U/mL RNase inhibitor) followed by RNA isolation with TRIzol. The small RNA libraries were prepared by NEBNext Small RNA Library Prep Kit (New England Biolabs) as the manufacturer described. In short, the adapters were ligated sequentially (3′ adapter, reverse transcription primer, 5′ adapter). First-strand cDNA synthesis was performed followed by PCR amplification with PCR index primer and PCR Master Mix to enrich the cDNA fragments. Finally, the PCR products were purified from PAGE gel and prepared for sequencing. Small

RNAs were sequenced using NovaSeq 6000 (Illumina) in paired-end mode. Reads were analyzed as described above (Bioinformatic analysis), from one set of the paired reads (R1) due to their short length.

## AGO2 immunoprecipitation

For AGO2 IP, three biological replicates of WT zygotes were injected with half-eGFP dsRNA and ~1,500 animals were frozen after 24 hours. IP of AGO2 was performed using custom antibodies, as previously described [80]. In brief, AGO2 antibodies (GenScript, USA) and rabbit IgG antibodies (Merck Millipore, I5006-10MG) were added to PBS-washed Protein A Magnetic Beads (MedChem Express, USA) and incubated overnight at 4 °C in rotation. Samples were homogenized with in lysis buffer [25 mM Tris-HCl (pH 7.4), 150 mM KCl, 25 mM EDTA, 0.5% NP-40, 1 mM DTT, Protease inhibitor cOmplete ULTRA tablets (Roche), Protease Inhibitor Cocktail Set III, EDTA-Free (Merck Millipore) and Murine RNAse inhibitor (New England Biolabs)], rotated and centrifuged at 4 °C for collection of the aqueous phase. The lysate was then added to pre-washed magnetic beads for pre-clearance, and the samples were rotated at 4 °C. Then, added to the antibody-bound beads and incubated in rotation at 4 °C before washing with wash buffer [50 mM Tris-HCl (pH 7.4), 300 mM NaCl, 5 mM MgCl$_2$,0.05% NP-40, Protease inhibitor cOmplete ULTRA tablets (Roche), Protease Inhibitor Cocktail Set III, EDTA-Free (Merck Millipore) and Murine RNAse inhibitor (New England Biolabs)]. The immunoprecipitated samples were kept in −80 °C until RNA extraction followed by sRNA library preparation.

## Trans-kingdom rapid affordable Purification of RISCs (TraPR)

For isolation of AGO-bound sRNAs, Trans-kingdom rapid affordable Purification of RISCs (TraPR) kit was used (Lexogen, Austria) [47]. Three biological replicates of frozen half-eGFP dsRNA-injected or uninjected WT animal samples (24 hours old 600−900 embryos) were processed according to manufacturer's protocol. Extracted sRNAs were kept in −80 °C until library preparation.

## Western blot

Animals were mechanically homogenized in the following lysis buffer: 50 mM Tris-HCl (pH 7.4), 150 mM KCl, 0.5% NP-40, 10% glycerol, protease inhibitor cOmplete ULTRA tablets (Roche) and Protease Inhibitor Cocktail Set III, EDTA-Free (Merck Millipore), or alternatively HALT protease inhibitor (Thermo Fisher Scientific). Protease inhibitors were added fresh just before use. After 2 hours rotation in 4 °C, the samples were centrifuged at 16,000 g for 15 min at 4 °C and supernatant was collected. Protein concentration was measured using Pierce BCA Protein Assay Kit (Thermo Fisher Scientific). Equal amounts of protein were run on 4–15% Mini-PROTEAN TGX Precast Protein Gel (Bio-Rad) followed by blotting to a polyvinylidene fluoride or nitrocellulose membrane (Bio-Rad). Next, the membrane was washed with TBST buffer (20 mM Tris pH 7.6, 150 mM NaCl, 0.1% Tween 20) and blocked with 5% skim milk (BD, USA) in TBST for 1 hour on the shaker at room temperature. Previously validated custom-made antibodies for *N. vectensis* GW182 [52], Dicer1 [50], AGO1 and AGO2 (Genscript) [31] or monoclonal mouse anti-GAPDH (Abcam, United Kingdom) serving as loading control were diluted 1:1,000 in TBST containing 5% BSA (MP Biomedicals, USA) and incubated with the membrane in a sealed sterile plastic bag at 4 °C overnight. The membrane was washed three times with TBST for 10 min and incubated for 1 h with 1:10,000 diluted peroxidase-conjugated anti-mouse, anti-rabbit or anti-Guinea pig antibody (Jackson ImmunoResearch, USA) in 5% skim milk in TBST. Finally, the membrane was washed three times with TBST, and detection was performed with the Clarity ECL kit (Bio-Rad) according to the manufacturer's instructions and visualized with a CCD camera of the Odyssey Fc imaging system (Li-COR Biosciences, USA). Size determination was carried out by simultaneously running Precision Plus Protein Dual Color Protein Ladder (Bio-Rad) and scanning at 700 nm wavelength. Band intensities were measured using the ImageStudio software (Li-COR). Band intensities were normalized according to the darkest band in each membrane then the normalized values of investigated proteins were divided by normalized values of GAPDH.

## FHV-eGFP and mCherry mRNA generation and injection

The sequence for recombinant FHV RNA1 coding RdRp fused to eGFP was ordered from Synbio Technologies based on previous publications [37,81]. The mCherry mRNA template was ordered as gBlock gene fragments (Integrated DNA Technologies) and in vitro transcribed as previously described [80]. For both constructs, the complete sequence including untranslated regions was amplified using Q5 High-Fidelity DNA Polymerase (New England Biolabs) and gene-specific primers introducing a T7 promoter and a Kozak sequence upstream of the coding region (primer sequences are available in S2 Data). In vitro transcription was conducted with HiScribe T7 mRNA Kit with CleanCap Reagent AG (New England Biolabs) following the manufacturer's instructions. The resulting RNA was purified using the RNA Clean and Concentrator-25 kit (Zymo Research) and eluted in 33 µl of RNase-free water. RNA concentration was quantified with the Qubit RNA Broad Range Assay Kit and a Qubit Fluorometer (Thermo Fisher Scientific). Polyadenylation was carried out using *Escherichia coli* Poly(A) Polymerase (New England Biolabs) for 30 min at 37 °C. The final mRNA products were purified with the RNA Clean and Concentrator-5 kit (Zymo Research) and eluted in 10 µl of RNase-free water. RNA integrity was verified on a 1.5% agarose gel after denaturation at 95 °C for 2 min in a thermocycler with a heated lid, cooling to 22 °C, and mixing with formamide (Merck Millipore) at a 1:3 ratio. The mRNA was stored in −80 °C until microinjection. FHV-eGFP mRNA (3,788 bp) was injected to *actin::eGFP* zygotes at concentration of 40 ng/µl, and mCherry mRNA (1,046 bp) was injected at 11.05 ng/µl to account for molarity. All injection mixes included dextran Alexa Fluor 594 at 50 ng/µl (Thermo Fisher Scientific, USA) as fluorescent tracer.

## DNA extraction

To detect potential reverse transcription of injected dsRNA, WT zygotes were injected with dsRNA and collected 72 hpi. Genomic DNA was extracted using an extraction buffer consisting of 10 mM Tris (pH 8), 150 mM EDTA (pH 8), 0.5% SDS, and 100 µg/ml Proteinase K (Thermo Fisher Scientific), added freshly before use. Samples were incubated in 1 ml of extraction buffer at 50 °C until complete tissue degradation, followed by an additional 30-min incubation at 37 °C with 3 µl of Monarch RNase A (New England Biolabs) to remove residual RNA. DNA was purified by sequential organic extraction steps: 1 ml of basic phenol (pH 7.5) was added to each sample, followed by centrifugation at 15,000 rpm for 3 min; the aqueous phase was retained. This was followed by extraction with an equal volume of phenol:chloroform:isoamyl alcohol (25:24:1) under the same centrifugation conditions, and again the upper phase was collected. A final extraction with chloroform was performed in the same manner. To precipitate DNA, 100 µl of 3 M Na-acetate (pH 5.2) and 1 ml of ice-cold ethanol were added and centrifuged for 10 min at 4 °C. The resulting DNA pellet was washed with 1 ml of 70% ice-cold ethanol, centrifuged again under the same conditions, air-dried, and dissolved in 50 µl of nuclease-free molecular-grade water. DNA samples were diluted 1:1 with nuclease-free water prior to PCR analysis.

## PCR

PCR was performed using Q5 High-Fidelity DNA Polymerase (New England Biolabs) with 35 cycles for 100 ng cDNA and 30 cycles for 0.3 µl diluted genomic DNA. Specific primers were used to amplify eGFP sequence and an endogenous gene (RLRa) as positive control (available in S2 Data). Product sizes were validated on 1.5% agarose gel with 1 kb Plus DNA Ladder (New England Biolabs).

## Microscopy

Fluorescence of eGFP protein was detected by SMZ18 stereomicroscope (Nikon) connected to an Intensilight fiber illumination fluorescent system (Nikon). Images were captured by DS-Qi2 SLR camera (Nikon) and were analyzed and processed with NIS-Elements Imaging Software (Nikon).

## Fluorescence analysis

Fluorescence intensity was measured using ImageJ software [82]. Between 20 and 30 embryos were selected per picture in 3–6 biological replicates per treatment and mean intensity values were calculated. Background intensity was subtracted and overlapping pixels or pixels not containing embryos were removed beforehand.

## Statistical analysis

Statistical analysis of small RNA read counts was done with one-way ANOVA with Tukey's HSD post-hoc test. For read counts of half-eGFP and full-length-eGFP two-tailed Student $t$ test was used. For comparison of AGO2 immunoprecipitated and IgG read counts and TraPR isolated dsRNA-injected and uninjected treatments two-tailed Student $t$ test was used. Comparison of eGFP transcript levels was conducted with one-tailed Student $t$ test, by comparing ΔCt values between groups. For ΔCt values between groups injected with FHV-GFP or mCherry mRNA, two-tailed Student $t$ test was used. Comparison of western blot band intensities was done with two-tailed Student $t$ test or Mann–Whitney–Wilcoxon Test for non-normal data by comparing normalized band intensities of the poly(I:C) compared to NaCl injected samples. For comparison of mean fluorescence intensity between treatments, averages of 20–30 measurements were tested using two-tailed Student $t$ test. For selection of embryos in each picture, blinding was used. Normality of the data and homogeneity of variance was validated before every statistical analysis. A $p$-value <0.05 was considered statistically significant. All experiments included at least three biological replicates, and three technical replicates were added for RT-qPCR. The tests were performed in Rstudio 2021.09.0.

## Supporting information

**S1 Fig. Read counts of primary siRNAs mapping to eGFP normalized by RPM. (A)** Number of reads (RPM) mapping to eGFP sequence, from embryos injected with eGFP dsRNA (24 hpi). *Actin::eGFP* zygotes were injected with either dsRNA or mock injection mix. WT zygotes were injected with dsRNA. Statistical significance is shown for pairwise comparison (Kruskal–Wallis rank sum test, $n = 4$ biological replicates, $p$-value = 0.02, pairwise comparison with Wilcoxon rank sum exact test and FDR correction, $p$-values = 0.04, 0.04, 0.68). **(B)** Size distribution of reads mapping to eGFP sequence corresponding to the samples in (A). **(C)** Number of reads (RPM) mapping to eGFP sequence following treatment of RNA with RppH enzyme or mock treatment. The treatment was performed on RNA extracted from *actin::eGFP* or WT embryos injected with dsRNA carrying eGFP sequence (24 hpi). Statistical significance is shown for pairwise comparison (one-way ANOVA, $n = 3$ biological replicates, $p$-value = 0.49, pairwise comparison with Tukey's HSD post-hoc test, $p$-values = 0.93, 0.81). **(D)** Number of reads (RPM) mapping to eGFP sequence, from embryos injected with either 3′ half- or full-length-dsRNA carrying eGFP sequence, or mock injection mix (24 hpi). Statistical significance is shown for pairwise comparison (two-tailed Student $t$ test, $n = 3$ biological replicates, $p$-value = 0.82). Bar graphs show the mean; error bars represent the standard error. The data underlying this Figure can be found in S1 Data.
(TIF)

**S2 Fig. eGFP knockdown in *actin::eGFP* embryos injected with dsRNA. (A)** Heterozygous *actin::eGFP* embryos showing dimmer eGFP fluorescence 24 hpi of dsRNA carrying eGFP sequence (left) compared to mock injection mix (right). Scale bars represent 500 µm. **(B)** eGFP transcript levels fold change measured by RT-qPCR, 24 hpi of dsRNA. Statistical significance is shown for pairwise comparison (one-tailed Student $t$ test, $n = 4$ biological replicates, $p$-value = 0.02). Box plot indicates the median and interquartile range, with whiskers showing the minimum and maximum values. The data underlying this Figure can be found in S1 Data.
(TIF)

**S3 Fig. siRNAs show no phasing or nucleotide bias. (A, B)** Radar plot showing the average distribution of 20–22 nt siRNAs across 20 phasing registers along the entire eGFP sense and antisense strands. The radial distance represents

the mean percentage of reads within each register, averaged from three biological replicates of dsRNA-injected *actin::eGFP* (A) or WT (B) zygotes. **(C, D)** Radar plot showing the average distribution of 20–22 nt siRNAs across 21 phasing registers along the entire eGFP sense and antisense strands. The radial distance represents the mean percentage of reads within each register, averaged from three biological replicates of dsRNA-injected *actin::eGFP* (C) or WT (D) zygotes. **(E, F)** Representation of sequence composition of total 20 (E) or 21 (F) nt eGFP-mapping reads, from three biological replicates of dsRNA-injected *actin::eGFP* zygotes. **(G, H)** Representation of sequence composition of unique 20 (G) or 21 (H) nt eGFP-mapping reads, from three biological replicates of dsRNA-injected *actin::eGFP* zygotes. Sequence logos were generated using WebLogo (78). The data underlying this Figure can be found in S1 Data.
(TIF)

**S4 Fig. eGFP dsRNA-derived siRNAs are loaded into AGO2. (A)** Number of reads (RPM) mapping to eGFP sequence, recovered from AGO2 immunoprecipitated, or IgG antibody as negative control, samples of WT embryos injected with dsRNA carrying eGFP sequence (24 hpi). Statistical significance is shown for pairwise comparison (two-tailed Student *t* test, $n = 3$ biological replicates, p-value = 0.006). **(B)** Size distribution of reads mapping to eGFP sequence corresponding to the samples in (A). **(C)** Number of reads (RPM) mapping to eGFP sequence, recovered from RISC-isolated samples of WT embryos injected with dsRNA carrying eGFP sequence (24 hpi) or uninjected as negative control (24 hours old). Statistical significance is shown for pairwise comparison (two-tailed Student *t* test, $n = 3$ biological replicates, *p*-value < 0.001). **(D)** Size distribution of reads mapping to eGFP sequence corresponding to the samples in (C). **(E)** Number of reads (RPM) mapping to eGFP sequence compared to miRNA reads, recovered from AGO2 IP samples of WT embryos injected with dsRNA carrying eGFP sequence (24 hpi). Statistical significance is shown for pairwise comparison (two-tailed Student t *t*est, $n = 3$ biological replicates, *p*-value = 0.01). **(F)** Number of reads (RPM) mapping to eGFP sequence compared to miRNA reads, recovered from RISC-isolated samples of WT embryos injected with dsRNA carrying eGFP sequence (24 hpi). Statistical significance is shown for pairwise comparison (two-tailed Student t *t*est, $n = 3$ biological replicates, *p*-value = 0.02). Bar graphs show the mean; error bars represent the standard deviation. Box plots indicate the median and interquartile range, with whiskers showing the minimum and maximum values. The data underlying this Figure can be found in S1 Data.
(TIF)

**S5 Fig. eGFP knockdown and siRNAs production in *actin::eGFP* embryos injected with half-dsRNA. (A)** Heterozygous *actin::eGFP* embryos showing dimmer eGFP fluorescence 24 hpi of 3′ half (left) or full length dsRNA carrying eGFP sequence (middle) compared to injection of mock injection mix (right). Scale bars represent 500 µm. **(B)** Normalized number of reads mapping to eGFP sequence, from embryos injected with either 3′ half or full dsRNA carrying eGFP sequence, or mock injection mix (24 hpi). Statistical significance is shown for pairwise comparison (two-tailed Student *t* test, $n = 3$ biological replicates, *p*-value = 0.82). **(C)** Size distribution of reads mapping to eGFP sequence corresponding to the samples in (B). Bar graphs show the mean; error bars represent the standard deviation in (B) and standard error in (C). The data underlying this Figure can be found in S1 Data.
(TIF)

**S6 Fig. Injected dsRNA is not reverse transcribed and is stable after injection. (A)** No eGFP amplicons are seen when amplified from genomic DNA extracted at 72 hpi of WT animals with eGFP dsRNA. Uninjected WT animals serve as negative control ($n = 3$ biological replicates). DNA ladder is 1 kb Plus DNA Ladder in all panels. **(B)** Amplification of endogenous gene as positive control to (A) shows clear bands (n = 3 biological replicates). **(C)** Amplification of eGFP was successful when amplified from cDNA transcribed at 24 and 72 hpi of WT animals with eGFP dsRNA ($n = 3$ biological replicates). **(D)** Amplification of endogenous gene as positive control to (C) shows clear bands ($n = 3$ biological replicates).
(TIF)

**S7 Fig. miRNA levels are largely unaffected by dsRNA injection. (A)** Normalized total number of miRNAs reads from embryos injected with dsRNA (24 hpi). *Actin::eGFP* zygotes were injected with either dsRNA or mock injection mix. WT zygotes were injected with dsRNA. Statistical significance is shown for pairwise comparison (one-way ANOVA, *p*-value = 0.708, *n* = 3 biological replicates, pairwise comparisons with Tukey's HSD post-hoc test, *p*-values = 0.68, 0.91, 0.89). **(B, C)** Differentially expressed miRNAs in *actin::eGFP* (B) and WT (C) embryos following injection of dsRNA (24 hpi). Box plot indicates the median and interquartile range, with whiskers showing the minimum and maximum values. The data underlying this Figure can be found in S1 Data.
(TIF)

**S8 Fig. *Dicer 1* and *AGO2* are enriched in immune cells.** Dot plot showing the percentage of cells expressing each gene (dot size) and the mean normalized expression level (color intensity) across annotated cell clusters from single-cell RNA-seq data.
(TIF)

**S1 Raw Images. Raw images for blots and gels.**
(TIF)

**S1 Data. Numerical source data and statistical details for figures.** This file contains numerical data underlying the figures displayed in this study. Each sheet contains the numerical data for the respective panels. The dataset includes ΔCt, ΔΔCt, and fold change values for all RT-qPCRs; mean intensity values of eGFP measured from injected *actin::eGFP* animals; raw and normalized read counts of eGFP and genome-mapped reads from small RNA libraries generated from dsRNA-injected animals, as well as raw spikeins read counts when applicable; and normalized per-base read counts across the eGFP sequence used to determine read-mapping positions. It further includes raw and normalized band intensities from western blot replicates of poly(I:C) injected animals, and raw and normalized read counts of miRNAs from small RNA libraries generated from dsRNA injected animals, including AGOs isolation by TraPR method or AGO2 immunoprecipitation.
(XLSX)

**S2 Data. DNA sequences.** DNA sequences of dsRNAs and primers used in this study.
(XLSX)

## Acknowledgments

The authors thank Dr. Aissam Ikmi (European Molecular Biology Laboratory, Heidelberg, Germany) for the kind gift of *actin::eGFP* anemones and Prof. Shou-Wei Ding (University of California, Riverside, CA, USA) for generously sharing the sequence of recombinant Flock House Virus RNA1. The authors also thank Prof. Petr Svoboda (Academy of Sciences of the Czech Republic, Prague, Czech Republic) and Dr. Filip Horvat (Max Perutz Labs, Vienna, Austria) for their advice in the phasing analysis and Dr. Michal Bronstein and Mrs. Adi Turjeman (The Center for Genomic Technologies, The Hebrew University of Jerusalem, Israel) for their help with high-throughput sequencing.

## Author contributions

**Conceptualization:** Yehu Moran.

**Data curation:** Yael Admoni, Junchao Shi, Xudong Zhang.

**Formal analysis:** Yael Admoni, Magda Lewandowska, Junchao Shi, Xudong Zhang.

**Funding acquisition:** Yehu Moran.

**Investigation:** Yael Admoni, Magda Lewandowska, Reuven Aharoni, Xudong Zhang.

**Methodology:** Yael Admoni, Magda Lewandowska, Reuven Aharoni, Xudong Zhang, Qi Chen.

**Project administration:** Yehu Moran.

**Resources:** Qi Chen, Yehu Moran.

**Supervision:** Qi Chen, Yehu Moran.

**Validation:** Qi Chen.

**Visualization:** Yael Admoni, Junchao Shi.

**Writing – original draft:** Yael Admoni, Yehu Moran.

**Writing – review & editing:** Yael Admoni, Magda Lewandowska, Reuven Aharoni, Junchao Shi, Xudong Zhang, Qi Chen, Yehu Moran.

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
