## [Editor Report · Decision Letter 0]

10 Nov 2025

Dear Yehu,

Thank you for submitting your revised manuscript entitled "Characterization of RNA interference in the model cnidarian Nematostella vectensis reveals partial target silencing but lack of small RNA amplification" for consideration as a Short Report by PLOS Biology. Please accept my apologies for the delay in getting back to you as we consulted with an academic editor about the revised version.

Your manuscript has now been evaluated by the PLOS Biology editorial staff, as well as by an academic editor with relevant expertise, and I am writing to let you know that we would like to send your submission back out to the original reviewers.

Once your full submission is complete, your paper will undergo a series of checks in preparation for peer review. After your manuscript has passed the checks it will be sent out for review. To provide the metadata for your submission, please Login to Editorial Manager (https://www.editorialmanager.com/pbiology) within two working days, i.e. by Nov 12 2025 11:59PM.

Kind regards,

Richard

Richard Hodge, PhD

rhodge@plos.org

PLOS

---

## [Decision Letter · Decision Letter 1]

5 Dec 2025

Dear Yehu,

Thank you for your patience while we considered your revised manuscript "Characterization of RNA interference in the model cnidarian Nematostella vectensis reveals partial target silencing but lack of small RNA amplification" for publication as a Short Report at PLOS Biology. This revised version of your manuscript has been evaluated by the PLOS Biology editors, the Academic Editor and the original reviewers.

Based on the reviews, we are likely to accept this manuscript for publication, provided you satisfactorily address the remaining points raised by the reviewers. Reviewer's #2 and #3 both note that the conclusions regarding siRNA-mediated antiviral responses should be toned down given the additional experiments do not demonstrate this definitively. In addition, Reviewer #2 raises some concerns with the documentation and controls for the experiment using self-replicating RNA. After discussions with the Academic Editor, while we agree that this additional work would be helpful, we ask that the limitations of the experiment are clearly stated and caveated in the manuscript text.

In addition, please also make sure to address the following editorial and data-related requests that I have provided below (A-G):

(A) We would suggest a very minor edit to the title, as follows, to help with flow. Please ensure you change both the manuscript file and the online submission system, as they need to match for final acceptance:

"Characterization of RNA interference in the cnidarian Nematostella vectensis reveals partial target silencing but lack of small RNA amplification"

(B) You may be aware of the PLOS Data Policy, which requires that all data be made available without restriction: http://journals.plos.org/plosbiology/s/data-availability. For more information, please also see this editorial: http://dx.doi.org/10.1371/journal.pbio.1001797

-Supplementary files (e.g., excel). Please ensure that all data files are uploaded as 'Supporting Information' and are invariably referred to (in the manuscript, figure legends, and the Description field when uploading your files) using the following format verbatim: S1 Data, S2 Data, etc. Multiple panels of a single or even several figures can be included as multiple sheets in one excel file that is saved using exactly the following convention: S1_Data.xlsx (using an underscore).

-Deposition in a publicly available repository. Please also provide the accession code or a reviewer link so that we may view your data before publication.

Figure 1D-E, 2A-D, 3A-H, 4C-F, 4K-N, S1A-D, S2B, S3A-D, S4A-F, S5B-C, S7A-C, S8

(C) Please also ensure that each of the relevant figure legends in your manuscript include information on *WHERE THE UNDERLYING DATA CAN BE FOUND*, and ensure your supplemental data file/s has a legend.

(D) We require the original, uncropped and minimally adjusted images supporting all blot and gel results reported in the following Figures:

Figure 4G-J, S6A-D

We will require these files before a manuscript can be accepted so please prepare and upload them now. Please carefully read our guidelines for how to prepare and upload this data: https://journals.plos.org/plosbiology/s/figures#loc-blot-and-gel-reporting-requirements.

(E) Please ensure that your Data Statement in the submission system accurately describes where your data can be found and is in final format, as it will be published as written there.

(F) Per journal policy, if you have generated any custom code during the course of this investigation, please make it available without restrictions. Please ensure that the code is sufficiently well documented and reusable, and that your Data Statement in the Editorial Manager submission system accurately describes where your code can be found.

(G) Please ensure that you are using best practice for statistical reporting and data presentation. These are our guidelines https://journals.plos.org/plosbiology/s/best-practices-in-research-reporting#loc-statistical-reporting and a useful resource on data presentation https://journals.plos.org/plosbiology/article?id=10.1371/journal.pbio.1002128

- If you are reporting experiments where n ≤ 5, please plot each individual data point.

We expect to receive your revised manuscript within two weeks. However, if you need some time then please do let us know, as the editorial office will be closed during the upcoming Christmas holidays and as a result we would extend to early January.

*Published Peer Review History*

*Press*

Best wishes,

Richard

Richard Hodge, PhD

rhodge@plos.org

Reviewer remarks:

Reviewer #1: The authors have provided clear and comprehensive responses to all my comments and questions, which has strengthened the manuscript. I fully support its publication.

Reviewer #2: Admoni et al. have analyzed features of RNA interference in the cnidarian model Nematostella vectensis. dsRNA injection into zygotes was used to induce RNAi and analyze its efficiency and other features. Authors show that Nematostella has a simple RNAi pathway without siRNA amplification and propose that RNAi in Nematostella provides only a short term antiviral mechanism.

The results are important for understanding evolution of dsRNA response and RNAi in particular as it is closely linked to the evolution of innate antiviral immunity. The revised version of the manuscript has addressed most points raised previously and I appreciate authors' efforts to address features of small RNAs generated from injected long dsRNA. I have two major issues with the revised version, which should be addressed before publication. In the minimal version both concern text revision.

Major issues

1) Authors should be precise and avoid overinterpreting their results. It has been pointed out during the first review and the issue still persists while I find it unnecessary. Authors examine dsRNA response and demonstrate presence of the canonical RNAi pathway = sequence-specific mRNA degradation induced by long dsRNA. The work on dsRNA response in this model is significant and sufficient. The fact that viral infection cannot be interrogated directly cannot excuse the use of the term of "antiviral RNAi". It does a huge disservice to the paper. Authors should critically revise all instances, where testing dsRNA response is called antiviral immune response, antiviral RNAi, antiviral response.

2) What is evidence that FHV-eGFP mRNA is actually a self-replicating system generating dsRNA? This experiment (Fig. 4A-F) is the most questionable in the manuscript - 1) a proper control would not be mCherry mRNA but FHV-eGFP mRNA with an RdRP-inactivating mutation, 2) there is no evidence of antisense strand synthesis, dsRNA presence, siRNA etc. This experiment does not bring any positive contribution to the manuscript (In my view, it does the opposite). It should be either removed or properly overhauled. A self-replicating system would strengthen the work only if it would be well documented.

Minor issues

1) Abstract 1st sentence "RNA interference (RNAi) is a sequence-specific mRNA degradation mechanism, which serves as an antiviral pathway by processing viral double-stranded RNA (dsRNA) into short interfering RNAs (siRNAs), leading to virus destruction. Please, revise as it would be confusing to general audience - it should state that siRNAs are guides for the sequence-specific mRNA degradation and indicate that antiviral role is one of roles of RNAi.

2) Abstract and elsewhere - please, clearly distinguish between innate immunity from acquired immunity, otherwise a text like "(anemones) possess an immune response with more vertebrate characteristics than previously thought" becomes misleading.

3) Figure 2 legend in the text on page 8 starts with the panel G while Fig. 2 has panels A-D - perhaps some automatic numbering ruined that.

4) Please, include more detailed description of spikeins and normalization in the methods - at the moment, normalization to million of reads is more comprehensible/informative about abundance of small RNAs than the relative numbers from spikein normalization in main figures.

5) I disagree with authors interpretation that "The highly reproducible pattern of the mapping suggests that Dicer has a sequence preference for processing of the injected dsRNA, as seen in previous experiments in mammalian cells." Such a pattern is likely highly influenced by strand selection (mentioned in the next sentence) and thus might also arise if the processing would be entirely random. Since authors do not observe strong phasing, it is plausible that the reproducible mapping pattern largely reflects strand selection of stochastically processed dsRNA.

6) Is Dicer in Nematostella processive? i.e. uses ATP-dependent helicase domain to process dsRNA - for an innate immunity RNAi system one would expect that it would be. Authors should discuss the helicase domain and true processivity in the discussion, as this feature is important for placing their results into the context of other analyses of Dicer and its ability to process long dsRNA.

Reviewer #3: The revised manuscript presents new data supporting a weak and transient siRNA response in Nematostella vectensis triggered by either injected dsRNA or self-replicating RNA. While I acknowledge the biological and technical constraints of this model system, I strongly recommend that the authors temper their conclusions and reframe key statements to describe a "likely antiviral siRNA response" rather than stating it as definitive (see lines 41-42 and 441-442).

Additionally, I have several minor comments:

L. 24: Please review the definition of RNAi in the abstract. RNA interference isn't just an antiviral pathway.

L. 90: Should be AGO1 instead of AGO.

L. 143 and other p values: show only the two first integers.

Supplementary Figure 1C-D are not mentioned in the text.

L. 293-294: in flies or mosquitoes reverse transcription of viral RNA into viral DNA is necessary for amplification of the siRNA response but it's not a prerequisite to produce antiviral siRNAs. I suggest adding ref. 17 here.

---

## [Editor Report · Decision Letter 2]

18 Dec 2025

Dear Yehu,

On behalf of my colleagues and the Academic Editor, Rene Ketting, I am pleased to say that we can accept your manuscript for publication, provided you address any remaining formatting and reporting issues. These will be detailed in an email you should receive within 2-3 business days from our colleagues in the journal operations team; no action is required from you until then. Please note that we will not be able to formally accept your manuscript and schedule it for publication until you have completed any requested changes.

PRESS

Best wishes, 

Richard

Richard Hodge, PhD

rhodge@plos.org

PLOS
